# Bioelectrical Impedance Analysis and Mid-Upper Arm Muscle Circumference Can Be Used to Detect Low Muscle Mass in Clinical Practice

**DOI:** 10.3390/nu13072350

**Published:** 2021-07-09

**Authors:** Dorienke Gort-van Dijk, Linda B.M. Weerink, Milos Milovanovic, Jan-Willem Haveman, Patrick H.J. Hemmer, Gerard Dijkstra, Robert Lindeboom, Marjo J.E. Campmans-Kuijpers

**Affiliations:** 1Faculty of Medicine, University of Amsterdam/Amsterdam UMC, Master Evidence Based Practice in Health Care, Meibergdreef 9, 1105 AZ Amsterdam, The Netherlands; 2Department of Nutrition and Dietetics, University Medical Center Groningen, University of Groningen, Hanzeplein 1, 9713 GZ Groningen, The Netherlands; 3Department of Radiology and Surgery, University Medical Center Groningen, University of Groningen, Hanzeplein 1, 9713 GZ Groningen, The Netherlands; l.b.m.weerink@umcg.nl (L.B.M.W.); m.milovanovic@umcg.nl (M.M.); 4Department of Surgery, University Medical Center Groningen, University of Groningen, Hanzeplein 1, 9713 GZ Groningen, The Netherlands; j.w.haveman@umcg.nl (J.-W.H.); p.h.j.hemmer@umcg.nl (P.H.J.H.); 5Department of Gastroenterology, University Medical Center Groningen, University of Groningen, Hanzeplein 1, 9713 GZ Groningen, The Netherlands; gerard.dijkstra@umcg.nl (G.D.); m.j.e.campmans-kuijpers@umcg.nl (M.J.E.C.-K.); 6Department of Epidemiology and Data Science, Amsterdam UMC, Academic Medical Center, Meibergdreef 9, 1105 AZ Amsterdam, The Netherlands; r.lindeboom@amsterdamumc.nl

**Keywords:** body composition, anthropometry, muscle mass, sarcopenia, computed tomography, bioelectrical impedance analysis, mid-upper arm muscle circumference, patient generated-subjective global assessment short form, advanced cancer

## Abstract

Identification of low muscle mass becomes increasingly relevant due to its prognostic value in cancer patients. In clinical practice, mid-upper arm muscle circumference (MAMC) and bioelectrical impedance analysis (BIA) are often used to assess muscle mass. For muscle-mass assessment, computed tomography (CT) is considered as reference standard. We investigated concordance between CT, BIA, and MAMC, diagnostic accuracy of MAMC, and BIA to detect low muscle mass and their relation with the clinical outcome malnutrition provided with the Patient-Generated Subjective Global Assessment Short Form (PG-SGA SF). This cross-sectional study included adult patients with advanced esophageal and gastrointestinal cancer. BIA, MAMC, and PG-SGA-SF were performed. Routine CT-scans were used to quantify psoas muscle index (PMI) and skeletal muscle area. Good concordance was found between CT_PMI_ and both BIA_FFMI (fat free mass index)_ (ICC 0.73), and BIA_ASMI (appendicular skeletal muscle index)_ (ICC 0.69) but not with MAMC (ICC 0.37). BIA_FFMI_ (94%), BIA_ASMI_ (86%), and MAMC (86%) showed high specificity but low sensitivity. PG-SGA-SF modestly correlated with all muscle-mass measures (ranging from −0.17 to −0.43). Of all patients with low muscle mass, 62% were also classified with a PG-SGA-SF score of ≥4 points. Although CT remains the first choice, since both BIA and MAMC are easy to perform by dieticians, they have the potential to be used to detect low muscle mass in clinical practice.

## 1. Introduction

Among patients with advanced cancer, malnutrition is common [1,2] and accompanied by weight loss [3] and loss of fat free mass (FFM) [4]. Poor nutritional status, particularly a low FFM, at the start of treatment is associated with a rising incidence of chemotherapy-related toxicity, a prolonged length of hospital stay after surgery, an increased risk of postoperative complications, and mortality [5,6,7,8,9]. Therefore, identifying, prevention, and treatment of malnutrition might offer an opportunity to enhance quality of patient care, improve clinical outcomes, and reduce healthcare costs [10,11].

There is growing evidence that early and evidence-based dietary counselling leads to improved dietary intake (energy and protein), body weight, nutritional status, enhanced oncologic and quality of life outcomes, and reduced complications [12,13,14,15,16]. In the treatment of advanced cancer, assessment of muscle status serves as an important indicator for clinicians to decide whether or not to proceed with an intended cancer treatment [17,18]. Identification of low muscle mass becomes increasingly relevant due to its prognostic value in cancer patients [19]. However, low muscle mass is not routinely recognized in current practice, since the assessment of nutritional status is mainly based on overall weight loss or decreased body mass index (BMI) alone, which does not differentiate fat mass from muscle mass [20,21].

For the assessment of clinical outcomes, the Patient Generated-Subjective Global Assessment (PG-SGA) is an established tool to screen for malnutrition in oncology settings [22] and is adapted for this population by Ottery [23] using the SGA tool developed by Detsky [24]. The PG-SGA consists of two parts: part one, known as the PG-SGA Short Form (PG-SGA SF), contains a self-assessment on the patient’s weight, food intake, symptoms, and functional ability. Part two assesses the metabolic demands of the disease and its impact on nutritional requirements and includes a physical examination of muscle mass and metabolic abnormalities by a trained clinician. It is a rapid, cost-effective, and feasible tool, which can easily be implemented in clinical settings, but its relation with measures of muscle mass is less well known [25].

For an accurate examination of nutritional status and to support clinical decision making, measurements of body composition, in particular muscle mass, are essential. There are various methods of body composition assessment. Presently, computed tomography (CT) and magnetic resonance imaging (MRI) are considered the reference standards to assess body composition in research [26]. CT provides precise segmentations of individual muscle and adipose tissue components. The total skeletal muscle mass based on CT estimation of the skeletal muscle area at the third lumbar vertebra (L3) is strongly correlated with total body skeletal muscle mass [27,28]. However, routine use of CT images for detection of low muscle mass in clinical dietary practice is limited. Additionally, dieticians are generally not trained to assess these scans [29,30].

Less demanding anthropometric methods, which can be performed by dieticians, include the mid upper-arm muscle circumference (MAMC) and bioelectrical impedance analysis (BIA). BIA measures the opposition (impedance) to the flow of an electrical current passed through the body, whereby impedance and its components, resistance, and reactance are quantitatively related to body water and, hence, fat-free mass [31]. Both MAMC and BIA are quick, easy, non-invasive, and non-expensive [32,33]. The importance of accurate assessment of muscle mass and identification of at-risk patients is increasingly recognized in clinical practice [34].

Currently, two studies on muscle loss in patients with advanced cancer are being conducted at the University Medical Centre Groningen (UMCG). In both studies, measurements of body composition parameters, including muscle mass, are measured with CT, BIA, and MAMC. Additionally, nutritional status is assessed with the PG-SGA.

The aim of this study was to examine concordance between CT-, BIA-, and MAMC- measures of muscle mass. The second aim was to investigate diagnostic accuracy of MAMC and BIA to detect low muscle mass. Furthermore, we examined their relation to a clinical outcome as assessed with the PG-SGA SF. Finally, we examined how conventional PG-SGA SF cut-offs for malnutrition relate to the diagnosis of low muscle mass as measured with CT.

## 2. Materials and Methods

### 2.1. Study Sample

Data was obtained from two studies on muscle loss in adult patients with advanced cancer. The first study concerns the study “*Sarcopenia Preventing in Oesophagectomy Trial (SPOT)*” (Trial NL6179 (NTR6326)). This intervention study aims to investigate whether a goal directed nutritional support can reduce sarcopenia (muscle loss) and the incidence of anastomotic leakage and pneumonia and prolong survival in oncological patients undergoing chemoradiotherapy and esophagectomy. The second study concerns the study “*Sarcopenia and Malabsorption after HIPEC (SMal-HIPEC)*” (Trial NL5961 (NTR6327)). The SMal-HIPEC study is a prospective observational study on high-risk surgical oncological patients with peritonitis carcinomatosa undergoing cytoreductive surgery and a HIPEC procedure. The aim is to investigate predictive factors for sarcopenia and malabsorption and identify prognostic factors for the incidence of anastomotic leakage, pneumonia, length of stay, readmission rate, and survival. Both studies are ongoing since 2017 at the UMCG. Therefore, this cross-sectional study included two samples. Both studies adhered to the Helsinki declaration, and METC approval was obtained [35]. Written informed consent was obtained for all participants. For the current study, patients were excluded if they wore a pacemaker or electronic implantable devices, since that precludes bio-electrical impedance.

### 2.2. Procedures

#### 2.2.1. Assessment of Muscle Mass by Computed Tomography Scan

CT scans completed for initial cancer staging and routine diagnostic purposes were used to quantify psoas muscle and skeletal muscle areas. Axial CT images obtained on the level of the third lumbar vertebra (L3) were used to calculate muscle surface. The most cranial slice, clearly displaying both transverse processes of the third lumbar vertebra, was used. All CT images were obtained 60–70 s after the administration of intravenous iodized contrast media. The thinnest available slice thickness, generally 1 mm, was used. Muscle mass was determined by manual outlining of the total cross-sectional area of both psoas muscles (total psoas area, TPA, mm^2^) and the total cross-sectional skeletal muscle area (SMA, cm^2^). The maximum surface area of both psoas muscles combined was used in the analysis. The measurements were performed by an experienced radiologist in training (LBMW) and a trained researcher (MM) with the use of imaging analysis software (Aquarius Intuition, Terarecon Inc., Foster City, CA, USA). These assessors were blinded to the outcome of the other measurements. The intra-observer correlation was 0.98 for SMA and 0.96 for TPA. The inter-observer correlation was 0.96 for SMA and 0.91 for TPA. TPA was normalized for patient height to calculate the psoas muscle index (PMI) in mm^2^/m^2^ and SMA was normalized for height to calculate the skeletal muscle index (SMI) in cm^2^/m^2^. Low muscle mass was defined using the sex-specific lowest quartile for the PMI as the cut-off point, as described in a study of Ozawa et al. [36]. For SMI, low muscle mass was defined using the cut-off point, according to Martin et al. [4]: SMI < 41 cm^2^/m^2^ for women; SMI < 43 cm^2^/m^2^ for men with BMI < 25 kg/m^2^; and SMI 53 cm^2^/m^2^ for men with BMI ≥ 25 kg/m^2^. To keep accuracy as high as possible, CT scans that performed closest to the data of the other muscle mass assessed on the baseline were used.

#### 2.2.2. Assessment of Muscle Mass by Bio-Electrical Impedance Analysis

BIA was analyzed in patients with a portable, multi-frequency eight-point Seca medical Body Composition Analyzer mBCA525. Waist circumference was measured from the mid-level between the iliac crest and the lowest rib. Patients were instructed to lie supine on a (hospital) bed with no limbs touching the body during the actual measurement. Fat mass and fat free mass were calculated using the regression equation of the manufacturer [37]. The fat-free-mass index (FFMI) was considered the primary BIA measure. The FFMI was calculated by the following equation: (total body weight [kg]—fat mass [kg])/height [m^2^]. Low muscle mass was defined using the cut-off point according to Schutz et al. [38]: the FFMI less than the 10th percentile of an age-matched and sex-matched Caucasian population was considered for the diagnosis of low muscle mass. As an additional parameter, the appendicular skeletal muscle index (ASMI) calculated from the appendicular skeletal muscle mass (ASMM) was used. The ASMI was calculated using Sergi’s formula [39] by the following equation: ASMM [kg] = −3.964 + (0.227 × RI) + (0.095 × weight) + (1.384 × sex) + (0.064 × Xc). Thereafter, the ASMI was normalized for height to calculate the ASMI in kg/m^2^_._ Low muscle mass was defined using the cut-off point, according to Gould et al. [40]: the ASMI < 7.0 kg/m^2^ for men and the ASMI < 5.5 kg/m^2^ for women were considered for the diagnosis of low muscle mass.

#### 2.2.3. Assessment of Muscle Mass by Mid Upper-Arm Muscle Circumference

Weight and height were recorded according to the standard methods from which BMI was calculated (weight [kg]/height [m^2^]). Mid upper-arm circumference was measured at the midpoint between the tip of the shoulder and the elbow on the non-dominant side of the body using a flexible tape, read to the nearest 0.1 cm. Triceps skin fold thickness measurements were performed with a Harpenden skinfold caliper (British indicators Ltd., St Albans, Herts, UK), read to the nearest 0.2 cm. Mid upper-arm muscle circumference (MAMC) was calculated: MAMC [mm] = mid upper-arm circumference [mm]—(3.14 × triceps skin fold thickness [mm]). Low muscle mass was defined using the cut-off point according to Frisancho [41]: MAMC less than the 10th percentile of an age-matched and sex-matched Caucasian population was considered for the diagnosis of low muscle mass.

#### 2.2.4. Assessment of Nutritional Status by PG-SGA SF

The PG-SGA is a validated questionnaire to assess nutritional status [22,23,42]. The score of the complete PG-SGA ranges between 0–52 points. The PG-SGA SF, which can be completed by the patients themselves in 5 min, contains the first four boxes, with scores ranging between 0–36 points [43]. Box 1: weight (history) and acute weight changes (scores 0–5); Box 2: food intake over the past month (scores 0–4); Box 3: nutrition impact symptoms experienced over the previous 2 weeks (scores 0–23); and Box 4: activity and functioning over the previous month (scores 0–4). The total score of these four boxes determine the level of nutritional risk according to prespecified nutritional triage recommendations: scores 0–1 (no intervention, regular reassessment), scores 2–3 (patient and family education by a dietician or nurse as indicated by symptoms), scores 4–8 (intervention by a dietician in conjunction with nurse or physician, as indicated by symptoms), and scores ≥9 (critical need for improved symptom management and/or nutritional intervention options) [23].

In accordance with the PG-SGA triage system, a PG-SGA SF total score of 0–3 points was categorized as low risk, 4–8 points as medium risk, and ≥9 points as high risk for malnutrition [44,45]. Cut-offs used for analysis were based on this triage system and set as follows: medium/high risk (≥4 points) and high risk (≥9 points).

#### 2.2.5. Assessment of Dietary Intake

Dietary intake was assessed by 24 h recall. Calories and proteins were then calculated using nutritional calculation software (Evry).

### 2.3. Statistical Analysis

Descriptive data were reported as mean and standard deviation (SD) if normally distributed, otherwise in median and interquartile range (IQR, 25–75%). Normality of data was checked by visual assessment and using Shapiro–Wilk tests. In case of violation of the normality assumption, nonparametric statistics were used. After converting to z-scores, the Intraclass Correlation Coefficient (model1, one-way agreement) was calculated to determine the concordance between CT, BIA, and MAMC. ICCs less than 0.40 were considered as ‘poor’, between 0.40–0.59 as ‘fair’, between 0.60–0.74 as ‘good’, and between 0.75–1.00 as ‘excellent’ [46]. In addition, differences between measures of CT-, BIA-, and MAMC were visualized with Bland–Altman plots, and limits of agreement were calculated. Cross tabulation was used to determine sensitivity, specificity, and the diagnostic odds ratio (DOR) for BIA and MAMC measures in identifying low muscle mass using CT_PMI_ as the reference standard. DOR is the ratio of the odds of a positive test result on BIA or MAMC in patients with low muscle mass relative to the odds in patients without low muscle mass [47].

Spearman’s rank correlation coefficients were used to examine the relation between the PG-SGA SF and the muscle mass measurements. To evaluate how the conventional PG-SGA SF cut-offs for malnutrition and diagnoses of low muscle relate, we compared the median PG-SGA score and numbers of malnutrition cases among low and normal muscle mass subjects. All statistical analysis were performed by using R version 3.5.3.

## 3. Results

A total of 60 patients were initially included in the study. In four patients, BIA measurement could not be performed due to malfunction of the device. In addition, another seven patients were excluded because their CT-scans were unsuitable for muscle mass analysis. A total of 49 patients were included in the final analysis (Figure 1). Median age was 62.0 (IQR, 56.0–70.0) years and 53.1% were men. In 26.5% of the patients, BMI was ≥30 kg/m^2^. Most patients (91.8%) had no limitations in their activities or functioning and were ambulatory with fairly normal activities (PG-SGA SF Box 4 score 0 or 1). One patient (2.1%) was not feeling up to most things and was in bed or a chair less than half the day (PG-SGA SF Box 4 score 2), and three patients (6.1%) were only able to do little activity and spent most of the day in bed or a chair (PG-SGA SF Box 4 score 3). BIA, MAMC, and PG-SGA SF were measured on the same day; in one patient only BIA was measured 1 week later. Median time between performing CT and other measurements was 14.0 (IQR, 12.0–34.0) days (Table 1).

Table 2 shows outcomes of the muscle mass measurements derived from CT, BIA, and MAMC and outcomes of the PG-SGA SF. For the total sample, mean CT_SMI_ was 45.5 ± 13.4 cm^2^/m^2^. Mean CT_SMI_ was higher in women than men, respectively, 47.8 ± 14.3 cm^2^/m^2^ vs. 43.3 ± 12.4 cm^2^/m^2^ (*p* = 0.25). Median CT_PMI_ was 58.0 (IQR, 50.0–71.0) mm^2^/m^2^ and was higher in men 69.5 (IQR, 56.5–83.0) mm^2^/m^2^ than women 53.0 (IQR, 44.0–59.0) mm^2^/m^2^ (*p* <0.05). For the total sample, mean BIA_FFMI_ was 18.8 ± 2.8 kg/m^2^, mean BIA_ASMI_ was 7.0 ± 1.2 kg/m^2^, both higher in men than in women (both *p* < 0.05). Mean MAMC for the total sample was 25.2 ± 4.7 cm and 26.7 ± 4.9 cm for men and 23.4 ± 3.7 cm for women (*p* = 0.01). The median PG-SGA SF score was 3.0 (IQR, 0.0–7.0) for the total sample. Twenty patients (40.8%) had a score of 4 points or more and eight patients (16.3%) had a score of ≥9 points.

### 3.1. Concordance of Muscle Mass Measurements between CT, BIA, and MAMC

Concordance was calculated based on z-scores. Table 3 shows the concordance between CT-, BIA-, and MAMC- measurements of muscle mass. Concordance between CT_SMI_ and CT_PMI_ was poor (ICC −0.07), as was concordance between CT_SMI_ and measurements derived from BIA_FFMI_ (ICC −0.06), BIA_ASMI_ (ICC −0.06), and MAMC (ICC −0.01). Concordance with CT_PMI_ with both BIA_FFMI_ (ICC 0.73) and BIA_ASMI_ (ICC 0.69) were good. Concordance between CT_PMI_ and MAMC (ICC 0.37) was fair. MAMC showed good concordance with BIA_FFMI_ and BIA_ASMI_, with ICC 0.64 and ICC 0.71, respectively. Differences between methods with good concordance were plotted against the mean of the two measurements and are shown in Figure 2. The Bland−Altman analysis showed limits of agreement ranging from −1.45 to 1.45 z-score for CT_PMI_ (corresponding to deviations ranging from 7.4% and 92.6% of the total possible range of the measures) and BIA_FFMI._ for CT_PMI_ and BIA_ASMI_, limits of agreement ranging from −1.56 to 1.56 z-score (corresponding to deviations ranging from 5.9% and 94.1% of the total possible range of the measures). For BIA_FFMI_ and MAMC, limits of agreement ranged from −1.69 to 1.69 z-score (corresponding to deviations ranging from 4.6% and 95.5% of the total possible range of the measures). For BIA_ASMI_ and MAMC, limits of agreement ranged from −1.50 to 1.50 z-score (corresponding to deviations ranging from 6.7% and 93.3% of the total possible range of the measures).

### 3.2. Diagnostic Accuracy of BIA and MAMC to Identify Low Muscle Mass

Of the 49 study participants, 23 patients (46.9%, 15 men, 8 women) were identified as having low muscle mass based on CT_SMI_. Based on CT_PMI_, 13 patients (26.5%, 7 men, 6 women) were identified as having low muscle mass. Five patients (10.2%), three men and two women, were classified as having low muscle mass by BIA_FFMI_. Ten patients (20.4%), five men and five women, were classified as having low muscle mass by BIA_ASMI_. Furthermore, nine patients (18.4%), five men and four women, were classified with low muscle mass by MAMC. In Table 4, we present the accuracy of BIA and MAMC in identifying patients with low muscle mass. The outcomes illustrate to which extent false positive and false negative BIA and MAMC outcomes contribute to the mismatch with CT_PMI_. For identification of low muscle mass, both BIA_FFMI_, BIA_ASMI_, and MAMC had low sensitivity. Specificity was high: 94%, 86%, and 86%, respectively. DORs were 5.1 for BIA_FFMI_, 3.8 for BIA_ASMI_, and 2.8 for MAMC.

### 3.3. Relation between Muscle Mass Measurements with Clinical Outcome (PG-SGA SF)

PG-SGA SF showed modest negative correlations with CT_SMI_ (rho = −0.17, *p* = 0.246) and BIA_ASMI_ (rho = −0.43, *p* = 0.002). Meanwhile, correlations between PG-SGA SF and CT_PMI_, BIA_FFMI_, and MAMC were 0.32, 0,38, and 0.26, respectively.

### 3.4. Low Muscle Mass and PG-SGA SF Cut-Offs for Malnutrition

In Table 5, we present the numbers with medium and high risk of malnutrition (PG-SGA SF cut-offs) among patients with low and normal total muscle mass according to CT_PMI_. For patients with low muscle mass, the median PG-SGA SF score was 5 (IQR, 2.0–9.0) compared to 1.5 (IQR, 0.0–6.0) for patients with normal muscle mass. Of the 13 patients with low muscle mass, 62% were classified as medium/high risk of malnutrition compared to 33% for patients with normal muscle mass. Using the PG-SGA SF ≥9 points cut-off for malnutrition, these were 31% and 11%, respectively. Patients with low muscle mass were 3.2 times more likely to be at medium/high risk for malnutrition and 3.6 more likely to be at high risk for malnutrition.

## 4. Discussion

This study shows a high correlation between the z-scores for measuring muscle mass in surgical oncological patients for CT and BIA. Additionally, BIA and MAMC showed good concordance. However, Bland and Altman plots of z-score deviations were typically in the range of ±1.5, indicating large differences between muscle mass measures on individual levels, suggesting that BIA, MAMC, and CT cannot be used interchangeably. Using conventional cut-offs for low muscle mass for BIA_FFMI_, BIA_ASMI_, and MAMC, specificity was high, whereas sensitivity was low. Of all patients with low muscle mass, 62% were also classified with a PG-SGA SF score of ≥4 points, underscoring the importance of screening for muscle mass in clinical practice.

These findings correspond with results from previous studies on muscle mass measurements. A recent study by Looijaard et al. [48] on critically ill patients also showed significant correlations—ranging between 0.64–0.834—for different BIA-derived muscle mass equations and CT-derived measurements. Giusto et al. [49] found a rather weak correlation (0.48 for men and 0.18 for women) between MAMC with CT scan analysis of muscle mass in patients with liver cirrhosis. This corresponds to the correlation we found between MAMC and CT_PMI_.

Both CT_SMI_ and CT_PMI_ have demonstrated that they are applicable for the assessment of muscle mass in patients with advanced cancer [4,36]. Remarkably, in our study, only CT_PMI_ was related to BIA and MAMC assessment of muscle mass. Another notable finding was that women had higher muscle mass than men when measured with CT_SMI_ (*p* = 0.25), but not with CT_PMI_ (*p* < 0.05). This might be due to overrepresentation of men in the esophageal sample who were in worse nutritional status than the, mainly female, patients in the peritonitis carcinomatoses sample. Patients with esophageal cancer often experience nutritional intake problems at diagnosis [50]. However, it contradicts results of the other muscle mass measurements, where men overall showed higher muscle mass than women. The low correlation between CT_PMI_ and CT_SMI_ also suggests that these CT methods cannot be used interchangeably in our sample of oncologic patients. In line with our finding, a recent review and meta-analysis concluded that low psoas mass prior to surgery better predicts the development of postoperative complications than total skeletal muscle mass [51]. Another review also highlighted inconsistencies in current literature as to defining muscle mass parameters measured by a CT scan and emphasized the need for standardized protocols and definitions [52].

Early identification of (risk of developing) low muscle mass may lead to more timely nutritional support which, in turn, may benefit the prognosis of patients under treatment for cancer [53]. Therefore, more accurate and routine measurement of muscle mass in clinical practice is crucial for aligning appropriate interventions to prevent any further muscle loss. BIA and MAMC both underestimated the presence of low muscle mass in our study. This implies that neither of these two methods can match the precision of CT scans. As has been shown in previous research, transverse CT images on the level of the third lumbar vertebra (L3) strongly correlates with total body skeletal muscle area in patients with cancer [27,54], whereas MAMC only represents the arms and can only be determined indirectly by measuring mid upper arm circumference and triceps skinfold thickness. Fluid imbalance, often occurring in cancer patients, can result in an erroneous measurement of BIA [55]. However, both BIA and MAMC showed high specificity at the conventional cut-offs used in screening, which makes these instruments suitable for detecting low muscle mass in patients, in both clinical and primary care settings where assessment of muscle mass with CT scan is not feasible. Therefore, both BIA and MAMC might be well suited for routinely assessment of muscle mass, especially because they are widely available, non-expensive, and relatively easy to perform.

Among patients with low muscle mass, 62% were also classified with a PG-SGA SF score of ≥4 points, indicating medium/high risk for malnutrition. Since malnutrition is accompanied with loss of muscle mass, identification of patients at risk for malnutrition seems relevant in clinical practice to prevent further progression of low muscle mass [56]. Recently, a review of Deutz et al. also emphasized the relationship between low muscle mass and malnutrition and therefore the need for screening patients [57]. Nevertheless, our study demonstrated that the PG-SGA SF and muscle mass measurements with CT, BIA, and MAMC cannot be used interchangeably. However, CT image analysis effectively adds value to nutrition screening [58].

There are some limitations to consider regarding the body composition measurement techniques used in this study. Although BIA has extensively been validated as a body composition measurement tool, hydration status can affect resistance measured by BIA [59]. As fluid shifts are common in patients with cancer, BIA measurements may have overestimated fat-free mass and thus underestimated the presence of low muscle mass. In addition, MAMC measurements are subject to variability as observers need to be experienced in the measurement technique. Different observers carried out the MAMC measurements, which may have affected its reliability. Furthermore, only 49 subjects could be included. This may have limited our analysis and affected our results. We recommend the conduct of a study with a larger sample. The strength of this study was that all measurements, except CT, were performed on the same day, allowing a direct comparison between the clinical values of the available muscle mass measures.

In summary, the concordance between BIA and CT was high. For BIA_FFMI_, BIA_ASMI_, and MAMC, specificity was high, but sensitivity was low. CT remains the first choice in detecting low muscle mass in clinical practice. However, both BIA and MAMC could be used to detect low muscle mass in clinical practice. Since these tools are easy to perform by dieticians, they might be well suited for routine assessment of muscle mass in clinical practice.

## Figures and Tables

**Figure 1 nutrients-13-02350-f001:**
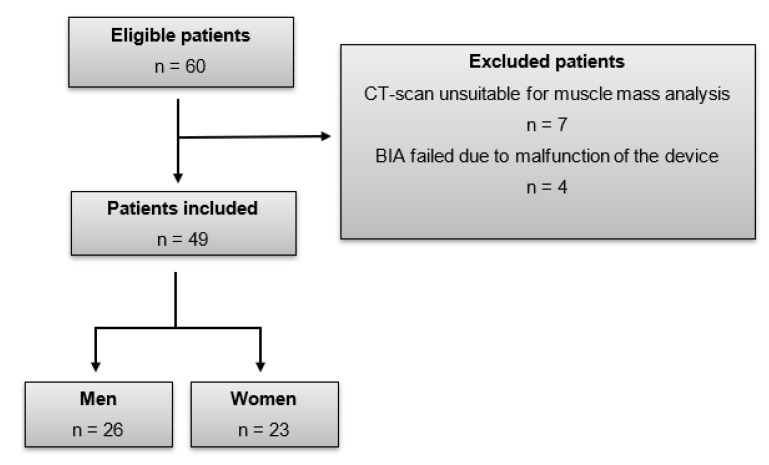
Flow chart of inclusion process.

**Figure 2 nutrients-13-02350-f002:**
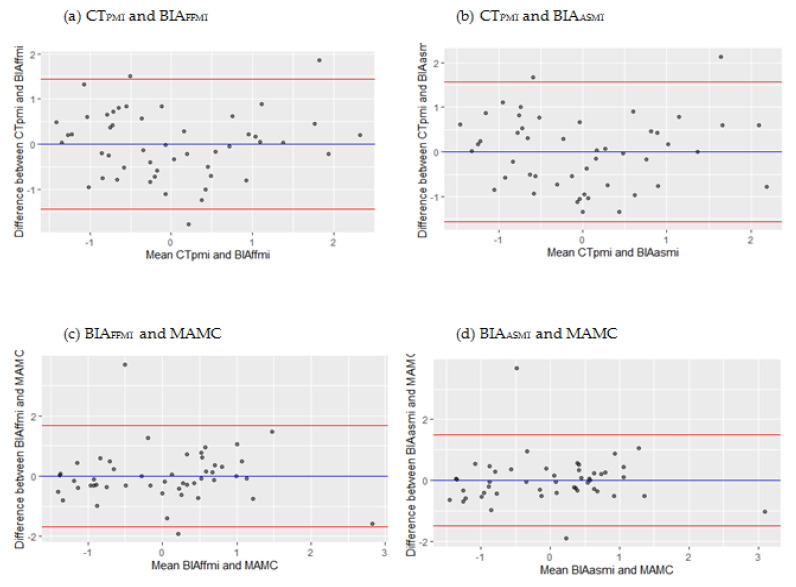
Bland−Altman plots showing absolute differences based on z-scores between muscle mass methods against the mean of the two measurements in the total sample.

**Table 1 nutrients-13-02350-t001:** Clinical characteristics of patients.

Characteristics	All Patients (*n* = 49)	Men (*n* = 26)	Women (*n* = 23)
Age (years; median, IQR)	62.0 (56.0–70.0)	65.0 (57.0–68.5)	62.0 (50.5–71.5)
Height (cm; mean (SD)	174 ± 8.5	179 ± 6.2	167 ± 6.4
Weight (kg; mean (SD)	80.8 ± 17.1	88.7 ± 15.3	71.9 ± 14.6
BMI (kg/m^2^; mean ± SD) ^a^	26.8 ± 5.0	27.7 ± 4.6	25.7 ± 5.4
Underweight (*n*, %)	1 (2.0)	0 (0.0)	1 (4.3)
Normal weight (*n*, %)	20 (40.8)	9 (34.6)	11 (47.8)
Overweight (*n*, %)	15 (30.6)	9 (34.6)	6 (26.1)
Obesity or obese (*n*, %)	13 (26.5)	8 (30.8)	5 (21.7)
Weight loss past month ^b^			
No weight loss (*n*, %)	35 (71.4)	18 (69.2)	17 (74.0)
0–5% weight loss (*n*, %)	9 (18.4)	6 (23.2)	3 (13.0)
5–10% weight loss (*n*, %)	2 (4.1)	1 (3.8)	1 (4.3)
>10% weight loss (*n*, %)	2 (4.1)	1 (3.8)	1 (4.3)
	Missing: 1		Missing: 1
Weight loss past 6 months ^b^			
No weight loss (*n*, %)	19 (38.8)	8 (30.8)	11 (47.8)
0–5% weight loss (*n*, %)	14 (28.6)	9 (34.6)	5 (21.7)
5–10% weight loss (*n*, %)	6 (12.2)	3 (11.5)	3 (13.0)
>10% weight loss (*n*, %)	5 (10.2)	3 (11.5)	2 (8.7)
	Missing: 5	Missing: 3	Missing: 2
Waist circumference (cm; mean ± SD) ^c^	99.8 ± 19.0	107 ± 20.1	91.5 ± 13.8
Underweight (*n*, %)	0 (0)	0 (0)	0 (0)
Healthy waist (*n*, %)	13 (26.5)	8 (30.8)	5 (21.7)
Overweight (*n*, %)	9 (18.4)	3 (11.5)	6 (26.1)
Obesity or obese (*n*, %)	27 (55.1)	15 (57.7)	12 (52.2)
Dietary intake (mean ± SD)			
Calorie intake (kcal/d)	1950 ± 461	2020 ± 399	1880 ± 516
Calorie intake (kcal/kg)	25.6 ± 9.3	23.4 ± 6.6	27.8 ± 11.0
Protein intake (gram/d)	88.0 ± 20.1	92.2 ± 20.2	83.7 ± 19.6
Protein intake (gram/kg)	1.2 ± 0.4	1.1 ± 0.3	1.2 ± 0.4
	Missing: 5	Missing: 4	Missing: 1
PAL (median, IQR)	1.8 (1.6–1.8)	1.8 (1.6–1.8)	1.8 (1.6–1.8)
Diagnosis (*n*, %)			
Esophageal cancer	24 (51)	18 (69.2)	6 (26)
Peritonitis Carcinomatosa	25 (49)	8 (30.8)	17 (74)
Time between CT and BIA, MAMC, PG-SGA SF (days; median, IQR)	14.0 (12.0–34.0)	13.0 (10.3–20.4)	19.0 (12.0–43.0)

BMI, body mass index; PAL, physical activity level (energy expenditure); CT, computed tomography; BIA, bioelectrical impedance analysis; MAMC, mid-upper arm muscle circumference; PG-SGA SF, patient generated-subjective global assessment short form; WHO, world health organization; NHG, Dutch general practitioner society. ^a^ WHO categories: underweight BMI < 18.5, normal 18.50–24.99, overweight ≥ 25, obesity or obese ≥ 30. ^b^ According to the PG-SGA SF. ^c^ NHG categories: underweight <79 cm (m) or <68 cm (w), healthy waist 79–94 cm (m) or 68–80 cm (w), overweight 94–102 cm (m) or 80–88 cm (w), obesity or obese ≥ 102 cm (m) or ≥88 cm (w).

**Table 2 nutrients-13-02350-t002:** Measurements of muscle mass and PG-SGA SF.

	All Patients (*n* = 49)	Men (*n* = 26)	Women (*n* = 23)
CT_SMI_ (mean ± SD)	45.5 ± 13.4	43.3 ± 12.4	47.8 ± 14.3
Low muscle mass (*n*, %)	23 (46.9)	15 (57.7)	8 (34.8)
	Missing: 1	Missing: 1	
CT_PMI_ (median, IQR)	58.0 (50.0–71.0)	69.5 (56.5–83.0)	53.0 (44.0–59.0)
Low muscle mass (*n*, %)	13 (26.5)	7 (26.9)	6 (26.1)
BIA_FFMI_ (mean ± SD)	18.8 ± 2.8	20.6 ± 2.1	16.7 ± 1.7
Low muscle mass (*n*, %)	5 (10.2)	3 (11.5)	2 (8.7)
BIA_ASMI_ (mean ± SD)	7.0 ± 1.2	7.7 ± 0.9	6.2 ± 0.9
Low muscle mass (*n*, %)	10 (20.4)	5 (19.2)	5 (21.7)
	Missing: 1		Missing: 1
MAMC (mean ± SD)	25.2 ± 4.7	26.7 ± 4.9	23.4 ± 3.7
Low muscle mass (*n*, %)	9 (18.4)	5 (19.2)	4 (17.4)
PG-SGA SF score (median, IQR)	3.0 (0.0–7.0)	1.0 (0.0–6.8)	3.0 (1.0–6.0)
≥4 points (*n*, %)	20 (40.8)	11 (42.3)	9 (39.1)
≥9 points (*n*, %)	8 (16.3)	4 (15.4)	4 (17.4)

CT, computed tomography; SMI, skeletal muscle index (cm^2^/m^2^); PMI, psoas muscle index (mm^2^/m^2^); BIA, bioelectrical impedance analysis; FFMI, fat free mass index (kg/m^2^); ASMI, appendicular skeletal muscle index (kg/m^2^); MAMC, mid-upper arm muscle circumference (cm); PG-SGA SF, patient generated-subjective global assessment short form (points).

**Table 3 nutrients-13-02350-t003:** Intraclass correlations for the concordance based on z-scores between muscle mass measures.

	CT_SMI_	CT_PMI_	BIA_FFMI_	BIA_ASMI_	MAMC
CT_SMI_	-	−0.07	−0.06	−0.07	−0.01
		(95%CI −0.35–0.21)	(95%CI −0.33–0.23)	(95%CI −0.34–0.22)	(95%CI −0.29–0.27)
CT_PMI_	-	-	0.73 ^a^	0.69 ^a^	0.37 ^a^
			(95%CI 0.57–0.84)	(95%CI 0.51–0.81)	(95%CI 0.1–0.59)
BIA_FFMI_	-	-	-	-	0.64 ^a^
					(95%CI 0.44–0.78)
BIA_ASMI_	-	-	-	-	0.71 ^a^
					(95%CI 0.54–0.83)
MAMC	-	-	-	-	-

CT, computed tomography; SMI, skeletal muscle index; PMI, psoas muscle index; BIA, bioelectrical impedance analysis; FFMI, fat free mass index; ASMI, appendicular skeletal muscle index; MAMC, mid-upper arm muscle circumference. Interpretation: less than 0.40 ‘poor’; between 0.40–0.59 ‘fair’; between 0.6–0.74 ‘good’; between 0.75–1.00 ‘excellent’ [46]. ^a^
*p* < 0.05.

**Table 4 nutrients-13-02350-t004:** Accuracy of BIA and MAMC in identifying patients with low muscle mass with CT_PMI_ as the reference method.

	True Positive (*n*)	False Positive (*n*)	False Negative (*n*)	True Negative (*n*)	Sensitivity	Specificity	DOR
BIA_FFMI_	3	2	10	34	23	94	5.1
BIA_ASMI_	5	5	8	30	38	86	3.8
MAMC	4	5	9	31	30	86	2.8

BIA, bioelectrical impedance analysis; FFMI, fat free mass index; ASMI, appendicular skeletal muscle index; MAMC, mid-upper arm muscle circumference; Sens, sensitivity; Spec, specificity; DOR, Diagnostic Odds Ratio.

**Table 5 nutrients-13-02350-t005:** Prevalence of malnutrition (PG-SGA SF cut-offs) among patients with low and normal muscle mass.

	Low Muscle Mass *	Normal Muscle Mass *
	(*n* = 13)	(*n* = 36)
PG-SGA SF score (median, IQR)	5 (2.0–9.0)	1.5 (0.0–6.0)
PG-SGA SF ≥4 points (*n*, %)	8 (62)	12 (33)
PG-SGA SF ≥9 points (*n*, %)	4 (31)	4 (11)

PG-SGA SF, patient-generated subjective global assessment short form. * According to CT_PMI_. All differences *p* > 0.05.

## Data Availability

The datasets of the current study are not publicly available, but are available from the corresponding author on reasonable request.

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
