# Peer review of "Bioelectrical Impedance Analysis and Mid-Upper Arm Muscle Circumference Can Be Used to Detect Low Muscle Mass in Clinical Practice"

_nutrients, 2021, doi:10.3390/nu13072350_

Round 1

Reviewer 1 Report

This is a cross-sectional study aimed to 1) examine concordance between Computed Tomography, Bioelectrical Impedance Analysis and mid upper-arm muscle circumference as measures of muscle mass, 2) investigate diagnostic accuracy of mid upper-arm muscle circumference Bioelectrical impedance analysis and to detect low muscle mass and 3) examine their relation with a clinical outcome (PG-SGA SF). I would like to mention the following changes/suggestions for being considered: Abstract-      In general, there are abbreviations that have not been introduced previously, and for the reader not specialized in the subject may give rise to doubt. This is the case of PG-SGA SF (line 31), PMI (line 32) and FFMI and ASMI (line 33). -      Review the text and correct errors such as the dots on lines 31 and 37. -      I do not find it meaningful to include "sarcopenia" among the keywords, because the definition of sarcopenia includes aspects such as loss of muscle strength and physical performance, which have not been included among the outcome variables. Introduction

It is well organized and the relevant aspects that justify the study are clearly explained.

 Methods-      Line 128. Please, describe how the determination of muscle mass was recorded using the cross-sectional area of both psoas muscles as the outcome variable. Please detail which outcome variable was used (mean or maximum of both). -      Line 114: “this cross-sectional study included two samples”. In this regard, my question is why was a sample calculation not performed for this derivative study? Since this is a cross-sectional study, this calculation is relevant.-      Please, provide information about how is collected the “dietary intake”? This information appears in table 1, but not in Methods section.  Results-      Line 216. Please, explain better what is de meaning of “BIA measurement could not be performed due to malfunction of the device”. Malfunction? Then, how many participants were measured by BIA? Table 1 detail missing 1 between women. -      I think it would be convenient to detail in how many cases fluid retention (lower limb edema, ascites, etc.) may have influenced the BIA measurement. As well as to argue that data and the results in the discussion section. Previous literature proposes calculating the phase angle from BIA data, as this is a more robust and fluid-independent outcome measure for calculating muscle mass from BIA. Perhaps reference 49 may be of help.-      In Tables 1 and 2, detail p-values when comparing the variables between men and women. Especially since the text comments "Mean CTSMI was higher in women than men…".   -      Please, check tables 1 and 2. There are missing parentheses in some data.-      In Table 1 I have not found the abbreviations for WHO and NHG.

Discussion -      Limitations have to include the low sample size.-      Lines 345-346: “Another notable finding was that women had higher muscle mass than men when measured with CTSMI, but not with CTPMI”. What is the p-value for this? This should be discussed as well as explaining in detail how the CTPMI variable has been recorded, as I have discussed in the Method section.-      The conclusion (lines 394-399) is not supported by the results, especially when it says "Both BIA and MAMC could be used to confirm…" From my point of view, the low concordance between CT (gold standard) and MAMC (proposed outcome measured) do not allow to say that the low mid upper-arm muscle circumference confirm low muscle mass (MAMC, ICC 0.37). The authors should be more cautious in their conclusions. It would better use the verb “detect” in place of “confirm”. This should also be taken into account in the abstract and title.-      Line 399. What is the meaning of this sentence? Why do authors consider the need for new cut-off points for BIA variables in cancer patients? References-      The references must be adapted to the style indicated in the template. For example, the comma should be removed from the last author's name, immediately before the title of the publication. Moreover, abbreviations of journal’s titles need revision.  Finally, it is convenient review English writing and punctuation marks. 

Reviewer 2 Report

A well-written manuscript describing an interesting study. There are some, mainly technical, points, however, I would like the authors to consider. Lines 73 and 74. Computed tomography etc do not require capitalization. Line 82. Since it is MUAC that is actually measured and then MAMC predicted you may wish to make this clear since prediction will inevitably have larger associated error. Lines 83 and 84 This sentence is very unclear. Do you mean "BIA measures the opposition (impedance) to the flow of an electrical current passed through the body whereby impedance and its components, resistance and reactance, are quantitatively related to body water and hence fat-free mass."? This is a more accurate and precise statement of what BIA actually is. Please re-word to make clear. Line 86. "Their" implies both methods. The accuracy of BIA for prediction of total muscle is questionable. It can be used to predict appendicular muscle (calibrated against MRI) but for the whole body BIA predicts FFM. This distinction will be important since your aim is to study concordance between methods (line 93). Line 148. No need for caps. More importantly, and related to the point above, you are now talking about FFM and note that this is the primary measure (line 149). This does not concur with the stated aims and my point above. Please ensure that your terminology is correct and consistent. However, you do then refer to ASMM but only as a secondary parameter. Line 155. Why the Sergi equation? There are others. I have no problem with you choosing this but please justify. This is important since BIA is an indirect predictive technique which is widely noted to require population-specific prediction equations. Line 198 and the output measures. The use of ICC (Lin's concordance may have been preferred) and B&A analysis are used to assess agreement, i.e. like with like. Although you are achieving this by conversion to z scores, the actual measurements are not like with like. (This is clear from the data in Table 2. (Note that the Table mixes means and medians, this makes comparisons difficult). Also the results (e.g., Table 3 and Fig 2) do not make it clear that the comparison is based on z scores. Please make the exact basis for comparisons clear at all times. Fig 2 and relevant text. The LOA are quoted as absolute values But they are large, approaching the range of values. I would suggest that you indicate this as expressing the LOA in % terms also. The Discussion is well written and not unduly speculative. The authors commendably, recognise the limitations of BIA.

Reviewer 3 Report

Title : Bioelectrical impedance analysis and mid-upper arm muscle circumference can be used to confirm low muscle mass in clinical practice.

It is acceptable to understand that the muscles obtained by abdominal computer tomography were highly positively correlated with the skeletal muscle volume of the whole body. According to Shen et al. [27] mentioned in the manuscript, the SM position measured in this study was L4-L5 instead of L3. It is recommended to clarify why the position of L4-L5 was not used to calculate SMA in this study.

The operating procedure of using CT to select SMA or Psoas muscle area range must rely on human expertise. In this study, the reliability-related data or explanations among operators were not stated, and the author is recommended to add it.

At present, the indirect body composition items that are widely accepted for comparison of the skeletal muscle mass of the limbs are FFM and appendicular lean soft tissue. The author indirectly discussed the relationship between MAMC and BIA, SMA and posas muscle area. In this study, why did the author not consider directly discussing the relationship between mid upper-arm muscle circumference (MAMC) and the above indicators? The author is recommended to clarify it.

Another serious problem is that there were only 49 subjects in this study. For the study of the correlation between MAMC and skeletal muscle mass, it is obvious that 49 subjects were few.

When discussing the estimation of low muscle mass or sarcopenia in the elderly, in addition to specific ethnic studies, a large proportion of patients with sarcopenia are underweight. In this study, there was only one subject who was underweight. It is obvious that there was a big deviation in the sampling sample for this question. Or it can be said that there were no (few) samples of low muscle mass in this study. It is recommended that the author consider in-depth whether the subject of the study was consistent.

In this study, BIAasmi, BIAFFMI and MAMC were used in Bland-Altman plots. The difference between BIA and MAMC can reach 2Z values. It showed that the LOA between MAMC and BIA did not meet the clinical application. The above results cannot support the conclusion of this study.

Round 2

Reviewer 1 Report

I thank the authors for responding in detail to the questions raised, and for improving the writing of the manuscript with the changes made.

Author Response

We would like to thank you for your response.

Reviewer 3 Report

Thank you for author’s detailed answers to my questions one by one. After the author's reply, there are still some questions or suggestions, as follows.

The number of subjects in this study was only 49. This limit on the number of people was also a serious problem for the value of this research in scientific research. In this study, there were only 26 male and 23 female subjects, respectively. It is also questioned whether the sampling of subjects was a normal distribution. In table1, the author made statistics on the distribution of obesity directly for the subjects of the plan. In such a small number of subjects, it is necessary to consider whether the sampling was a normal distribution. Table 2 also showed that whether it was CTSMI, CTPMI, BIAFFMI, BIAASMI, MAMC and other indicators for the condition of low muscle mass, there were more men than women in sampling. This was not the norm. It is suggested that the authors need to further discuss or improve the sampling problems in this study. 

Following the previous paragraph, the author tried to apply Z score standardization to the units of different methods in the Bland-Altman plot. Applied LOA to explore the consistency of various methods. But this is also as mentioned above, it is necessary to consider the number of people sampled and whether the sampling was a normal distribution. Whether the z-score conversion was appropriate in the case of only 49 people in this study and regardless of gender was a serious question. Most of the horizontal axis ranges in the four figures of Figure 2 were -2 Z-score to 2 Z-score. The difference on the vertical axis was about -1.5 z-score to 1.5 z-score. The LOA range was too wide, about 75%. I still think that the results of this research are difficult to apply to clinical applications.
